# Temperature Effect of van der Waals Epitaxial GaN Films on Pulse-Laser-Deposited 2D MoS_2_ Layer

**DOI:** 10.3390/nano11061406

**Published:** 2021-05-26

**Authors:** Iwan Susanto, Chi-Yu Tsai, Yen-Teng Ho, Ping-Yu Tsai, Ing-Song Yu

**Affiliations:** 1Department of Materials Science and Engineering, National Dong Hwa University, Hualien 97401, Taiwan; 810322001@gms.ndhu.edu.tw (I.S.); 610522022@gms.ndhu.edu.tw (C.-Y.T.); 2Department of Mechanical Engineering, Politeknik Negeri Jakarta, Depok 16424, Indonesia; 3International College of Semiconductor Technology, National Yang Ming Chiao Tung University, Hsinchu 30010, Taiwan; chia500@yahoo.com.tw; 4Department of Electronic Systems Research Division, Chung-Shan Institute of Science & Technology, Tao-Yan 325, Taiwan; ringo0911@gmail.com

**Keywords:** molybdenum disulfide, gallium nitride, van der Waals epitaxy, pulsed laser deposition, molecular beam epitaxy

## Abstract

Van der Waals epitaxial GaN thin films on c-sapphire substrates with a sp^2^-bonded two-dimensional (2D) MoS_2_ buffer layer, prepared by pulse laser deposition, were investigated. Low temperature plasma-assisted molecular beam epitaxy (MBE) was successfully employed for the deposition of uniform and ~5 nm GaN thin films on layered 2D MoS_2_ at different substrate temperatures of 500, 600 and 700 °C, respectively. The surface morphology, surface chemical composition, crystal microstructure, and optical properties of the GaN thin films were identified experimentally by using both in situ and ex situ characterizations. During the MBE growth with a higher substrate temperature, the increased surface migration of atoms contributed to a better formation of the GaN/MoS_2_ heteroepitaxial structure. Therefore, the crystallinity and optical properties of GaN thin films can obviously be enhanced via the high temperature growth. Likewise, the surface morphology of GaN films can achieve a smoother and more stable chemical composition. Finally, due to the van der Waals bonding, the exfoliation of the heterostructure GaN/MoS_2_ can also be conducted and investigated by transmission electron microscopy. The largest granular structure with good crystallinity of the GaN thin films can be observed in the case of the high-temperature growth at 700 °C.

## 1. Introduction

At present, advanced applications of optoelectronic and electronic devices have attracted considerable interest for utilization on flexible and wearable devices for displays, solar cells, and detectors [1,2]. The gallium nitride (GaN) compound semiconductor has been widely studied as a potential material to serve the new generation of optoelectronic and microelectronic devices [3,4]. The many excellent properties of GaN can provide chemical and mechanical stability, such as good electron-mobility, direct bandgap, thermal stability, and better conductivity [5]. Therefore, it has been utilized for the primary layers of heterostructure films for several applications, such as GaN-based high electron mobility transistors (HEMTs), light emitting diodes (LEDs), and UV photo-detectors [6,7,8]. However, in traditional epitaxial techniques, GaN films are grown on rigid substrates, which confines their applications because of the difficulty for flexible devices [9,10]. The van der Waals (vdW) epitaxial heterostructure has offered a promising method for integrating a variety of two-dimensional layered materials with unique characters and flexible functionalities [11]. The vdW bonds could also easily facilitate the layers to be exfoliated for transferring to another substrate [12], which will naturally give an advantage for flexible applications with electronic and optoelectronic devices in the future [13].

Graphene is the most popular 2D buffer layer for the vdW epitaxial growth of compound semiconductors, such as GaAs grown on Si with graphene [14], GaN epitaxial layers are grown on multilayer graphene at the temperature of 1000 °C by metal-organic chemical vapor deposition [15]. Besides graphene, transition metal dichalcogenides (TMDs) based on semiconductor materials have also attracted interest to be applied on 2D layer heterostructures with vdW epitaxial technique. However, TMDs cannot exist at high-temperature growth [16,17]. For the epitaxial growth, the lattice mismatch between the heterostructure layers is also an interesting issue. Among TMDs materials, molybdenum disulfide (MoS_2_) is considered as one of the promising materials with 2D vdW layer employed as a template for growing GaN films [18,19,20]. The pulsed laser deposition (PLD) system has proven to be a powerful tool for the deposition of large-area 2D materials [21]. Meanwhile, molecule beam epitaxy (MBE) facilitates a high accuracy controlling layer-by-layer growth at a lower growth temperature to achieve a high-quality GaN heterostructure [22]. Recently, the initial stages of GaN growth by MBE on graphene-covered SiO_2_ substrates with or without AlN buffer layers was investigated where the AlN buffer acted as seeds for GaN growth [23]. Until now, the GaN growth with the incorporation of techniques by the MBE system and the deposition of 2D MoS_2_ by the PLD technique has not been exploited in detail. For the production of higher quality GaN films, the suitable growth temperature is an essential parameter for the epitaxial growth [24,25,26]. Therefore, it would be an exciting topic to investigate these layers for getting a deep understanding of the heterostructure growth of GaN on 2D MoS_2_ layers.

In this study, GaN thin films were grown on 2D MoS_2_ by using plasma-assisted MBE via different growth temperatures. The buffer layers of 2D MoS_2_ on c-sapphire were deposited by PLD technique. The surface conditions of 2D MoS_2_ and vdW epitaxial GaN films were characterized by in situ monitoring equipment, including reflection high energy electron diffraction (RHEED) in the MBE system. The crystalline structure, morphology, surface chemical composition, and optical properties of the epitaxial layers are clarified in detail by ex situ characterizations. Due to the vdW bonding, the exfoliation of GaN was also conducted and observed by transmission electron microscopy. The temperature effects on the growth mechanism of vdW epitaxial GaN/MoS_2_ was investigated in the report.

## 2. Materials and Methods

Layered 2D MoS_2_ with the thickness of 2 nm was deposited on 2-inch c-sapphire substrate by using a PLD technique equipped with a KrF excimer laser (Coherent, California, USA) at substrate temperature of 800 °C under background pressure of 8 × 10^−6^ Torr [27]. Further, the ULVAC plasma-assisted MBE system (ULVAC, Kanagawa, Japan) was employed for growing the GaN thin films on 2D MoS_2_/c-sapphire substrates [24]. The GaN/MoS_2_ heterostructures were prepared at different substrate temperatures such as 500, 600 and 700 °C, named as T1, T2 and T3, respectively. The epitaxial growth parameters in detail are listed in Table 1, and the base pressure of MBE chamber was carried out at 6 × 10^−10^ Torr. Thus, the pressure of nitrogen plasma was fixed at 9.7 × 10^−5^ Torr, and the temperature of Ga K-cell was controlled at 800 °C in the beam equivalent pressure of 6 × 10^−8^ Torr. During the epitaxial growth, the surface conditions of MoS_2_ and GaN films was observed by RHEED operating at 20 kV as in situ characterization. After the growth process, the surface morphology and chemical composition of films were investigated using scanning electron microscopy (SEM, JSE-7000F, JEOL, Tokyo, Japan), atomic force microscopy (AFM, C3000, Nanosurf, Liestal, Switzerland) and X-ray photoelectron spectroscopy (XPS, K-Alpha, Thermo Scientific, Waltham, MA, USA). The vibration modes of molecules for the GaN and MoS_2_ were observed by Raman spectroscopy equipped with a laser wavelength of 532 nm and power intensity of 0.6 mW. The crystallography and thickness of GaN thin films are identified by high-resolution X-ray diffraction (HRXRD, D1, Bede Scientific Instruments, Durham, UK) and transmission electron microscopy (TEM, JEOL JEM-3010, Tokyo, Japan). The optical properties of GaN thin films, related to near band edge and yellow band emissions, were investigated by photoluminescence spectroscopy (PL) at room temperature using a UV laser of 266 nm. Finally, the exfoliation of GaN/MoS_2_ heterostructure in alcohol was conducted by using an ultrasonic cleaning machine for 10 min, and then it was transferred to square-mesh copper grids for TEM observation.

## 3. Results and Discussion

### 3.1. In Situ Characterizations by Reflection High Energy Electron Diffraction

Figure 1 provides in situ RHEED patterns of the substrate before MBE growth and three GaN films after the growth. Figure 1a shows the substrate’s RHEED pattern from MoS_2_ layer grown on c-sapphire after pre-nitridation for the growth. An in-plane 2D layer of MoS_2_ with crystalline structure has been demonstrated by the bright streaky line with a broader shape. All the MBE growth of GaN thin films was conducted on this 2D MoS_2_ layer. After the growth, Figure 1b–d shows the RHEED patterns for the GaN thin films deposited at the temperatures of 500, 600 and 700 °C, respectively. An elliptical pattern with a weak intensity is shown in Figure 1b for samples T1, which performs a low crystalline structure from a more amorphous-like construction of GaN surface. In Figure 1c of sample T2, the spots are connected with the ring pattern, suggesting the polycrystalline structure formed on the surface of GaN films. For the higher temperature growth, sample T3, the mixture patterns of spots and streaks are shown in Figure 1d, relating to the single crystal wurtzite GaN grown on the 2D MoS_2_ layer [28]. It indicates the improvement of surface quality for the GaN thin films, wherein a better crystallinity was generated as the substrate temperature increased. Further, the crystalline structure of the GaN thin films on 2D MoS_2_ will be confirmed in detail by using HRXRD and TEM.

### 3.2. The Observation of Atomic Force Microscopy

The study of the surface morphology to get direct evidence for the surface conditions on the substrate and GaN films was performed by AFM analysis. The surface texture was demonstrated in the scan area of 3 µm × 3 µm, as shown in Figure 2. Some particles were observed on the surface of the MoS_2_ and GaN films, which resulted in AFM image artifacts. Moreover, the root mean square (RMS) value is the root mean square average of the profile height deviations from the mean line, recorded within the evaluation length. The RMS value can present the surface roughness of the samples in the AFM measurements. The RMS value of PLD MoS_2_ on c-sapphire is 1.12 nm, and some MoS_2_ particles were observed on the surface, as shown in Figure 2a. After the growth, three GaN thin films have the RMS values of 12.91 nm (T1), 3.86 nm (T2), and 2.23 nm (T3), displayed in Figure 2b–d, respectively. The smaller RMS value elucidates the smoother surface condition established on the GaN films. The minimum RMS value was obtained from the sample T3, indicating the smoother of GaN surface realized using the growth temperature at 700 °C. On the contrary, T1 sample suffered a rougher surface compared to T2 and T3, proven by higher RMS value. It could appertain as the peaks and valleys constructed on GaN films’ surface at a low growth temperature. The peaks might also be related to GaN particles invented on the surface, which will be expressly confirmed by SEM observations.

### 3.3. The Observation of Scanning Electron Microscopy

Figure 3 shows the SEM observations for ensuring the detail of GaN morphology at different growth temperatures. The images of samples T1, T2 and T3 are displayed in Figure 3a–c, respectively. GaN has been grown on the substrate with some particles decorated on the surface. Those particles could be related to GaN or Ga droplets attending irregularly on the surface. In Figure 3a, they presented primarily with particle sizes of around 300 nm for the growth temperature of 500 °C. During the MBE growth, less surface diffusion of atoms at the lower growth temperature and the original MoS_2_ particles on the surface could result in the formation of GaN particles or Ga droplets on the surface. As the growth temperature increases, the particles become smaller, as shown in Figure 3b,c. The increasing substrate temperature could create higher energy to facilitate the Ga atoms more actively for desorbing, and Ga atoms could also migrate easily to react with N atoms to form layer-by layer GaN growth [29].

### 3.4. The Analysis of X-ray Photoelectron Spectroscopy

Furthermore, the de-convolution of XPS spectra in Ga-3d of the GaN films at various temperatures are shown in Figure 4a–c, respectively. Ga-3d XPS spectra were divided into three bonding elements (BE): Ga–N, Ga–Ga, and O–O. Peak position and percentage of BE are comprehensively tabulated in Table 2. These peak positions from all samples were generally located at the binding energy of 18 eV to 23 eV [30,31,32]. The first peak with lower binding energy came from Ga–Ga bonding, attributing with the Ga droplets formed on the GaN surface. Thus, O–O bonding at the highest binding energy was created due to GaN exposed the air; while, the highest peaks corresponded with Ga–N bonding for T1, T2, and T3 appeared at 19.91 eV, 20.03 eV and 19.89 eV. Their percentage values achieved up to 83.3%, 81.4%, and 92.7%, respectively. Consequently, the percentage ratio value between Ga–Ga and Ga–N (R_Ga–Ga/Ga–N_) are 13%, 11% and 5% for T1, T2, and T3, respectively. The sample grown at a higher temperature (T3) has a higher Ga–N bonding percentage and a lower R_Ga–Ga/Ga–N_ value based on XPS results. It indicates that sufficient heat energy during the MBE growth leads to a better Ga–N bonding of the films on 2D MoS_2_. According to the growth mechanism of GaN by MBE, the physicochemical processes involved in the incorporation of cations and anions. At the higher substrate temperature, the more desorption of excess Ga atoms and higher surface migration of atoms could make the better formation of GaN films. Therefore, the XPS result for the surface chemical composition of GaN performed the higher Ga–N bonding percentage of 92.7% for the sample T3.

### 3.5. The Analysis of Raman Spectroscopy

The Raman spectra of a substrate (S) and three GaN films (T1, T2 and T3) are displayed in Figure 5. Two peaks located at 382.6 and 407.4 cm^−1^ are related to E_2g_ and A_1g_ from the characteristic Raman modes, associated with Mo and S atoms from the basal plane [33]. After the MBE growth, these peaks were not present, because 2D MoS_2_ buffer layer were covered by GaN films. In comparison, four main peaks correspond to the sapphire substrate’s signal attended at 418.4, 449.4, 578, and 760 cm^−1^. However, the Raman modes of major GaN are located at 575 and 750 cm^−1^ near sapphire E_G_ modes [34]. So, the E_2_ transverse optical (TO) mode of GaN films was not easy to observe; even the intensity of the sapphire substrate’s signal was reduced after the GaN growth. It could be due to the very thinness of the GaN films. Fortunately, the GaN signal of A_1_ longitudinal optical (LO) mode at 735 cm^−1^ was observed in the shoulder of sapphire E_G_ modes at 750 cm^−1^. It indicates that the heterostructure GaN layers were grown on MoS_2_/sapphire substrate. With the increase in growth temperature, the peak became clearer and more comfortable to identify, especially for sample T3.

### 3.6. The Analysis of Photoluminescence Spectroscopy

The PL measurement was carried out at room temperature to investigate GaN films’ optical properties. The two main peaks presented at the near band edge (NBE), and the yellow band (YB) emissions were normalized from the PL spectra, as shown in Figure 6. The NBE peak, associating with excited electrons’ radiative transition, was located at 3.62 ± 0.01 eV [35]. There was no difference in band gap energy for different growth temperatures, but the relative intensity of NBE and YB emissions changed for three samples. A full width at half maximum (FWHM) of NBE peak can represent the optical property and crystallization GaN quality. The FWHM values of T1, T2, and T3 are 0.612, 0.464, and 0.396 eV, respectively. The higher intensity with a narrow peak corresponds with a higher optical property and crystallization quality constructed in GaN thin films. Therefore, sample T3 has a sharper and narrower peak than sample T1 and T2, indicating a better optical property of GaN films on 2D MoS_2_. On the other hand, YB emission came out in the range of 1.8 to 2.8 eV, attributed to defects structure formed in the GaN thin films [36]. The YB emissions of the GaN films could come from the shallow donor to the deep state transition, located at about 1 eV above the valence band maximum. The defects’ states might be due to cation vacancy (V_Ga_) or substitutional carbon (C_N_), and/or its complexes [37]. The higher intensity with a broad peak indicates more defects formed in the films. Based on the PL results, as shown in Figure 6, sample T3 has a smaller area of YB emission, indicating the minimum defects constructed in GaN films as the MBE growth temperature increases.

### 3.7. The Analysis of X-ray Diffraction

The HRXRD measurement performs the crystal microstructure of GaN films. Figure 7 displays the X-ray diffraction for the rocking curve profiles consisting of the original and fitting curve results. The FWHM in symmetric (0002) planes are 259 and 248 arcsec for sample T2 and T3, respectively. The higher curve intensity with a low FWHM value indicates the better crystalline quality created in the GaN films. However, the X-ray diffraction for the rocking curve profile of T1 was absent, suggesting an amorphous-like structure formed in the GaN films. It was consistent with the observations of RHEED patterns and PL spectra. The higher growth temperature, such as sample T3, could facilitate the perfection of microstructure built-in GaN films on 2D MoS_2_.

### 3.8. The Observation of Transmission Electron Microscopy

To identify the thickness of films, TEM observes in the cross-section of heterostructure GaN/MoS_2_/sapphire for T2 and T3 samples in Figure 8. The uniform GaN thin films have heteroepitaxial grown on the substrate. The GaN thickness of sample T2 and T3 are around 4 nm and 6 nm, respectively. The 2D MoS_2_ layer is about 2 nm. The higher growth temperature of 700 °C could make GaN film thicker. It indicates that sufficient thermal energy for the surface migration of Ga atoms in plane and inter plane for the MBE growth of GaN.

Finally, the GaN/MoS_2_ heterostructure exfoliation was conducted efficiently in ethanol solution by an ultrasonic cleaner. The solution was dried on the copper mesh for the TEM observation. Figure 9 shows the plan-view TEM images of three samples and their selective area electron diffraction (SAED) patterns. The GaN thin films were not continuous after the exfoliation process and in the form of granular structure in the TEM images. It could be due to the very-thin GaN films were destroyed by the ultrasonic vibration. As the growth temperature increases, we can find the bigger granular structure of GaN in Figure 9c. Meanwhile, the TEM-SAED patterns, as shown in Figure 9, performed the crystallinity of GaN films. For sample T1, the ring pattern showed polycrystalline structure of GaN in the small grain size. This ring diffraction pattern contributed from different crystalline planes of GaN. For sample T2, we found the transition of SAED patterns from rings to spots. The regular dotted pattern of sample T3 displayed the single crystal-like hexagonal GaN [38]. In summary, as the growth temperature increases, the crystallinity of GaN on 2D MoS_2_ can be enhanced according to the observation of the SAED patterns, which is consistent with the results of RHEED and HRXRD.

## 4. Conclusions

We experimentally investigated van der Waals epitaxial GaN thin films on the pulse-laser-deposited 2D MoS_2_ by plasma-assisted MBE technique. The epitaxial growth of GaN thin films were obtained as the substrate temperature increased from 500 to 700 °C. The MoS_2_/sapphire substrates and three GaN thin films grown at different temperatures were characterized by in situ RHEED, AFM, SEM, XPS, Raman, PL, XRD, and TEM, respectively. While the growth temperature increased, the RHEED pattern displayed the surface transitions from amorphous to crystal structures. The surface roughness decreased due to the elimination of GaN particles or Ga droplets by the observations of AFM and SEM. The better surface chemical composition of the GaN films was obtained by XPS measurement. Thus, the Raman spectra showed the LO A_1_ mode of GaN. The PL spectra demonstrated the optical properties’ enhancement with a sharper NBE emission and a lower yellow band emission from the defects. For the crystal microstructure analysis of the GaN films by HRXRD and TEM, the crystal structure of the GaN thin films on 2D MoS_2_ layers had a phase transition from polycrystalline to single-crystal structures as the growth temperature increased from 500 to 700 °C. The epitaxial growth of ~5 nm and uniform GaN thin films on layered 2D MoS_2_ was firstly confirmed by the technique of MBE. Due to the van der Waals bonding between GaN and 2D MoS_2_, the exfoliation of GaN films can be conducted easily, which will be transferable to different substrates for further applications in electronics and optoelectronics.

## Figures and Tables

**Figure 1 nanomaterials-11-01406-f001:**
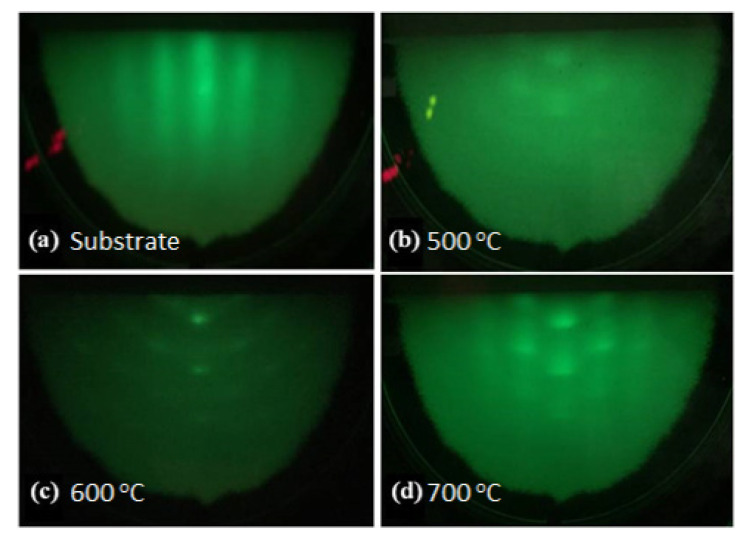
RHEED patterns of (**a**) substrate: 2D MoS_2_ on c-sapphire, (**b**) T1: GaN films grown on MoS_2_ at 500 °C, (**c**) T2: GaN films grown on MoS_2_ at 600 °C and (**d**) T3: GaN films grown on MoS_2_ at 700 °C.

**Figure 2 nanomaterials-11-01406-f002:**
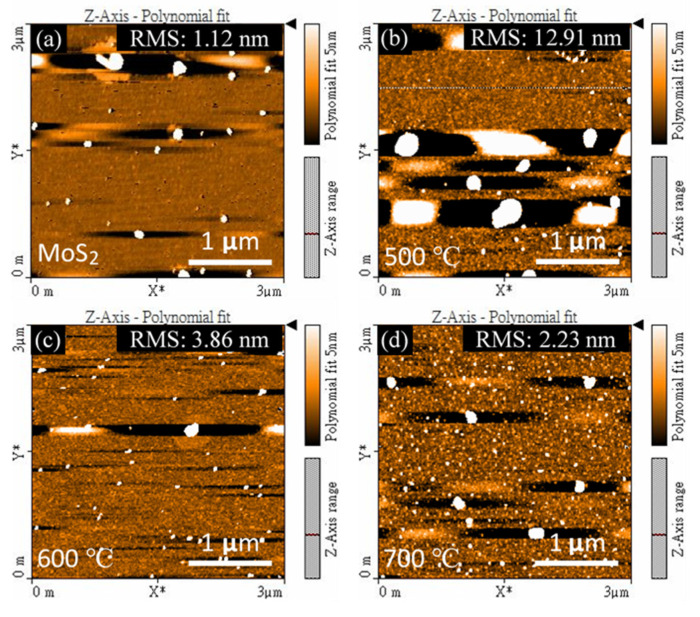
AFM images and RMS values of (**a**) substrate: 2D MoS_2_ on c-sapphire, (**b**) T1: GaN films grown on MoS_2_ at 500 °C, (**c**) T2: GaN films grown on MoS_2_ at 600 °C and (**d**) T3: GaN films grown on MoS_2_ at 700 °C.

**Figure 3 nanomaterials-11-01406-f003:**

SEM images of GaN films grown at different growth temperatures: (**a**) T1: 500 °C, (**b**) T2: 600 °C and (**c**) T3: 700 °C.

**Figure 4 nanomaterials-11-01406-f004:**
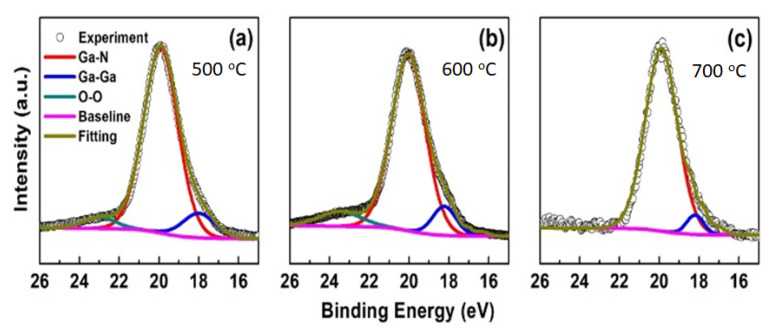
Ga-3d core-level spectra for GaN films: (**a**) T1: 500 °C, (**b**) T2: 600 °C, and (**c**) T3: 700 °C.

**Figure 5 nanomaterials-11-01406-f005:**
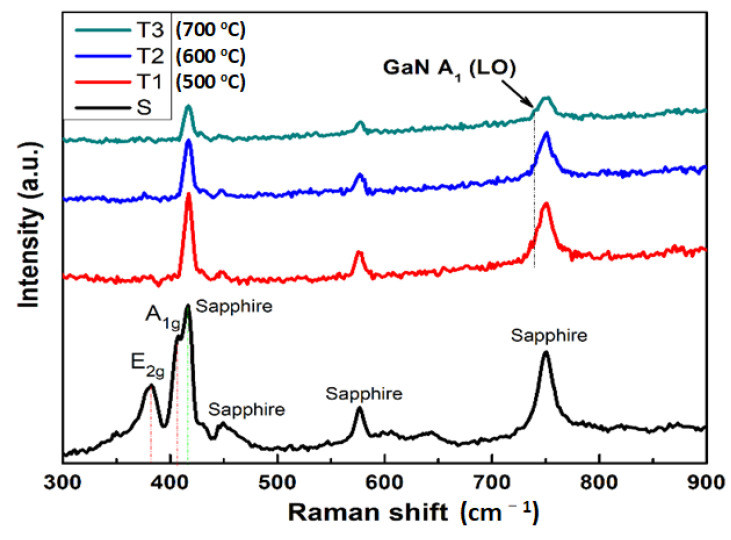
Raman spectra of 2D MoS_2_ on c-sapphire (S) and three GaN thin films at growth temperatures 500 °C (T1), 600 °C (T2), and 700 °C (T3).

**Figure 6 nanomaterials-11-01406-f006:**
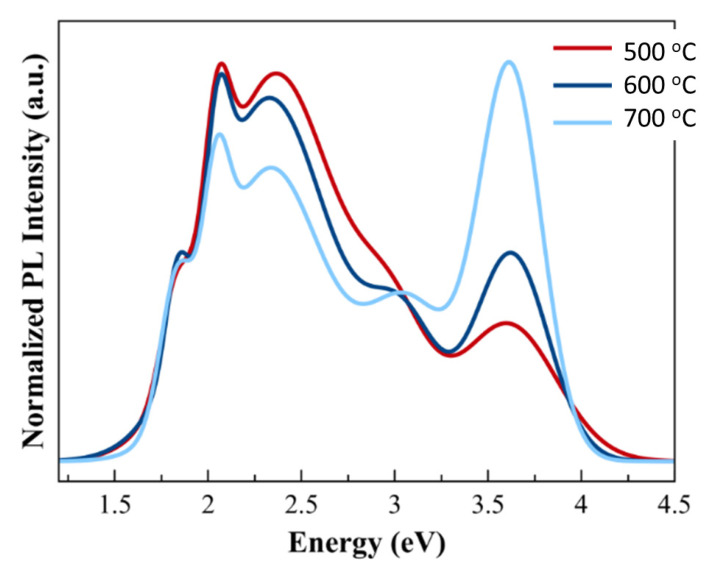
PL spectra at room temperature of GaN grown at various growth temperatures.

**Figure 7 nanomaterials-11-01406-f007:**
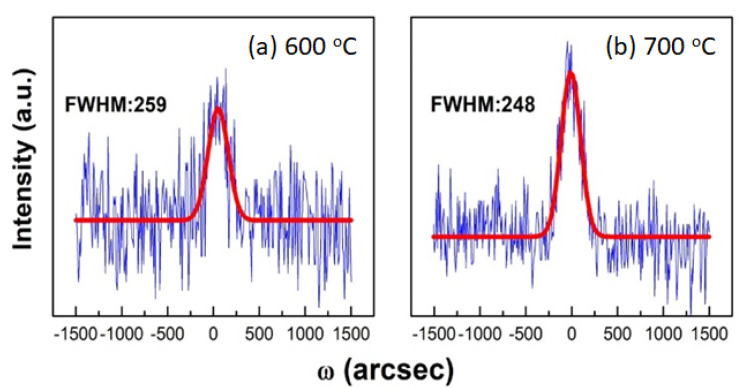
Rocking curve profiles and their fitting from HRXRD GaN (0002) at growth temperatures: (**a**) T2: 600 °C and (**b**) T3: 700 °C.

**Figure 8 nanomaterials-11-01406-f008:**
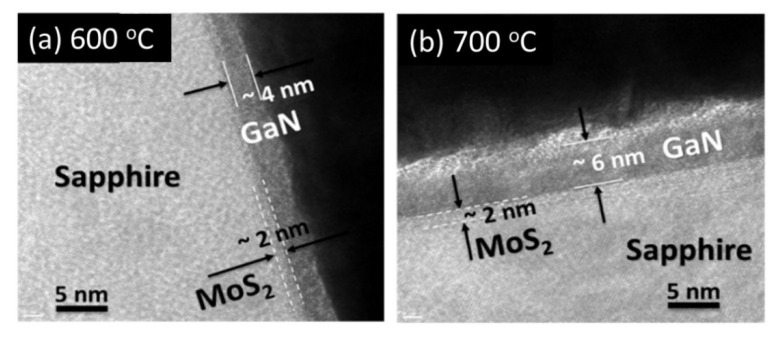
TEM images in the cross-section of GaN/MoS_2_/sapphire for (**a**) T2 and (**b**) T3 samples.

**Figure 9 nanomaterials-11-01406-f009:**
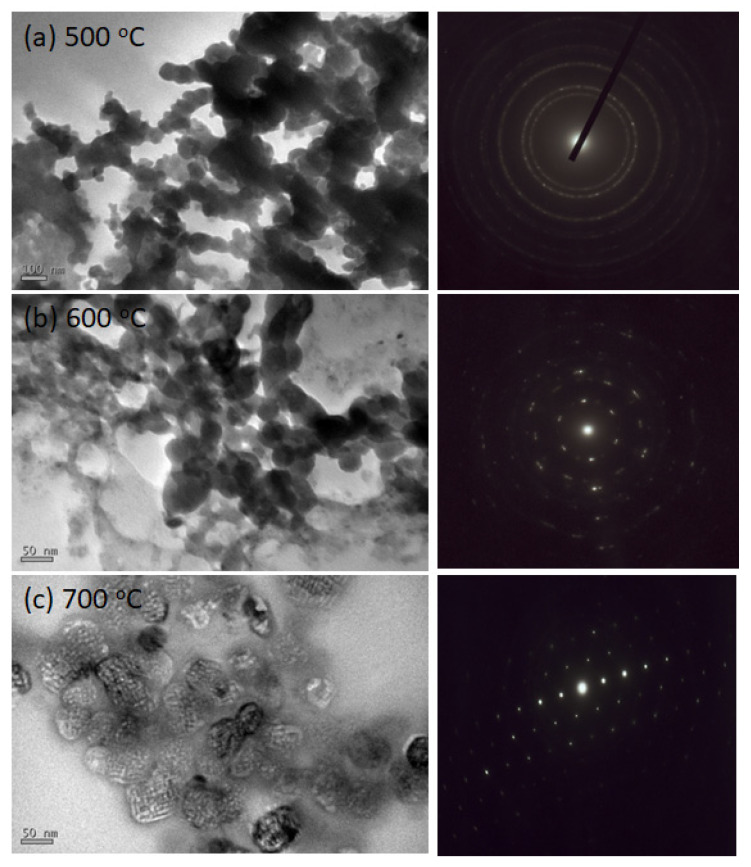
Plan-view TEM images of three samples and their electron diffraction patterns: (**a**) T1: 500 °C, (**b**) T2: 600 °C, and (**c**) T3: 700 °C.

**Table 1 nanomaterials-11-01406-t001:** Growth parameters of vdW epitaxial GaN thin films on layered 2D MoS_2_.

Sample	Substrate	Thermal Cleaning	Pre-Nitridation Treatment	GaN Growth Temperature	GaN Growth Time
T1	MoS_2_/c-sapphire	600 °C	600 °C	500 °C	20 min
T2		40 min	20 min	600 °C	
T3				700 °C	

**Table 2 nanomaterials-11-01406-t002:** Summary of a fitting parameter for Ga-3d core-level spectra (Figure 4). Peak position, percentage, and the ratio of the area under an individual peak.

Sample	Peak Position (eV)			Percentage (%)			R_Ga-Ga/Ga-N_ (%)
	Ga-N	Ga-Ga	O-O	Ga-N	Ga-Ga	O-O	
T1	19.91	18.08	22.98	83.3	11.1	5.6	13
T2	20.03	18.23	23.48	81.4	8.9	9.7	10
T3	19.89	18.28	23.15	92.7	5.1	2.2	5

## Data Availability

The data presented in this study are available on the request from the corresponding author.

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
