# Peer review of "Temperature Effect of van der Waals Epitaxial GaN Films on Pulse-Laser-Deposited 2D MoS2 Layer"

_nanomaterials, 2021, doi:10.3390/nano11061406_

Round 1

Reviewer 1 Report

In this work, GaN growth on 2D MoS2 layer was demonstrated. Three GaN samples were analyzed with various growth temperature. In addition, various analysis methodologies have been employed such as RHEED, AFM, PL, etc. Some issue should be resolved.

1. Growth of the GaN on MoS2 layer looks a novel technique. The reviewer is wondering if there is any particular improvement in the quality of GaN compared to the conventional methodology.

2. In order to show the clear quality comparison of GaN on various substrates such as MoS2 (this work), graphene, SiC, or bare sapphire, an additional table to compare material parameters is required for readers.

3. In page 5, author mentioned “It indicates that sufficient heat energy during the MBE growth leads to a better Ga-N bonding of the films on 2D MoS2”. An additional detail explanation of a correlation between growth temperature and Ga-N bonding is required.

4. In figure 6, what is estimated band gap energy from PL data for T1, T2, and T3? Is there any difference in band gap energy with different growth temperature?

5. In figure 6, yellow band (YB) emission shows separated two peaks at 1.8 and 2.8 eV. The reviewer is wondering if the author can describe the origin of those peaks.

 For all figures, labeling 500 oC, 600 oC, and 700 oC of growth temperature is recommended instead of T1, T2, T3.

Author Response

Dear Reviewer 1,

Thanks very much for your kindly help and insightful comments in order to make this manuscript more complete. We have replied to the reviewer’s comments with a point-by-point response below and given a revised manuscript. We hope that our correction could meet with your approval.

Reviewer 2 Report

The authors report an experimental study of the growth and characterisation of GaN thin layers on a MoS2 buffer layer on sapphire. The MoS2 buffer layer was grown via PLD and the GaN films using MBE at 3 different temperatures. The films are characterised by a wide range of techniques including AFM, SEM, XPS, Raman & PL spectroscopies, HRXRD and TEM/SAED, while the film growth is monitored in-situ using RHEED.

The research topic is an interesting and timely one which is likely to be of interest to the community and as such is original and novel. The main body of the work is devoted to presentation of experimental data and some relatively brief discussion of the likely origin of the systematic behaviour seems as a function of GaN film growth temperature. The main systematic behaviour seems to be an improvement in GaN crystalline quality with increasing growth temperature. However many of the explanations for the changes seen in individual characterisation measurements is rather speculative.

Overall I think the timeliness of the work means that it would be a useful addition to the research community to have it published, despite my reservations over some of the more speculative aspects of the discussion. Thus my recommendation is that the work be accepted after the authors have attend to the minor points I list below.

Technical points:

1 - Experimental methods - the laser used for Raman spectroscopy should be specified in terms of wavelength and intensity.

2 - in reference to the AFM data, the term RMS value is used. Do the authors mean the RMS roughness value (I presume so)? If so then this should be specifically stated.

3 - in reference to the AFM data in figure 2 (and especially 2(b)), the data close to the large particulates seems to show artifact behaviour (the black regions close to the particulates). The authors should comment on this and the possible reasons for it.

4 - in reference to the Raman data, given the weakness of non-resonant Raman scattering effects, it seems surprising that any signals are seen from either the MoS2 buffer layer or the GaN overlayers, given the thicknesses of these layers. Relating back to my first point above, what laser wavelength was used for the Raman spectroscopy, and was the scattering resonant in nature?

5 - for the HRXRD data, could the authors also supply the theta-2theta data in addition to the rocking curves on the (0002) GaN peak? This would be useful for the reader in terms of appreciating the overall diffraction behaviour of the system.

6 - in relation to the TEM-SAED data in figure 9, the authors claim that "the ring patterns showed the amorphous-like and polycrystalline structure" of the T1 sample. However, while the ring structure is clearly consistent with a polcrystalline deposit, the radial width of the rings does not seem to be much greater than the diffraction spot dimensions for the T2 and T3 samples and so I would question whether the attribution of an amorphous fraction to this deposit is justified, based on these data. The authors should discuss and justify this statement in a little more detail.

Author Response

Dear Reviewer 2,

Thanks very much for your kindly help and insightful comments in order to make this manuscript more complete. We have replied to the reviewer’s comments with a point-by-point response below and given a revised manuscript. We hope that our correction could meet with your approval.
